# Variation in aorta attenuation in contrast-enhanced CT and its implications for calcification thresholds

**Sven A. Holcombe**[1,2‡], **Steven R. Horbal**[1,2‡*], **Brian E. Ross**[1‡], **Edward Brown**[1‡], **Brian A. Derstine**[1‡], **Stewart C. Wang**[1,2‡]

1 Morphomics Analysis Group, University of Michigan, Ann Arbor, MI, United States of America,
2 Department of Surgery, University of Michigan, Ann Arbor, MI, United States of America

☯ These authors contributed equally to this work.
‡ SAH, SRH, BER, EB, BAD and SCW also contributed equally to this work.
* shorbal@med.umich.edu

**Data Availability Statement:** Dataset utilized is included in the paper and its Supporting information file.

## Abstract

### Background

CT contrast media improves vessel visualization but can also confound calcification measurements. We evaluated variance in aorta attenuation from varied contrast-enhancement scans, and quantified expected plaque detection errors when thresholding for calcification.

### Methods

We measured aorta attenuation (AoHU) in central vessel regions from 10K abdominal CT scans and report AoHU relationships to contrast phase (non-contrast, arterial, venous, delayed), demographic variables (age, sex, weight), body location, and scan slice thickness. We also report expected plaque segmentation false-negative errors (plaque pixels misidentified as non-plaque pixels) and false-positive errors (vessel pixels falsely identified as plaque), comparing a uniform thresholding approach and a dynamic approach based on local mean/SD aorta attenuation.

### Results

Females had higher AoHU than males in contrast-enhanced scans by 65/22/20 HU for arterial/venous/delayed phases ($p < 0.001$) but not in non-contrast scans ($p > 0.05$). Weight was negatively correlated with AoHU by 2.3HU/10kg but other predictors explained only small portions of intra-cohort variance ($R^2 < 0.1$ in contrast-enhanced scans). Average AoHU differed by contrast phase, but considerable overlap was seen between distributions. Increasing uniform plaque thresholds from 130HU to 200HU/300HU/400HU produces respective false-negative plaque content losses of 35%/60%/75% from all scans with corresponding false-positive errors in arterial-phase scans of 95%/60%/15%. Dynamic segmentation at 3SD above mean AoHU reduces false-positive errors to 0.13% and false-negative errors to 8%, 25%, and 70% in delayed, venous, and arterial scans, respectively.

**Funding:** The author(s) received no specific funding for this work.

**Competing interests:** The authors have declared that no competing interests exist.

## Conclusion

CT contrast produces heterogeneous aortic enhancements not readily determined by demographic or scan protocol factors. Uniform CT thresholds for calcified plaques incur high rates of pixel classification errors in contrast-enhanced scans which can be minimized using dynamic thresholds based on local aorta attenuation. Care should be taken to address these errors and sex-based biases in baseline attenuation when designing automatic calcification detection algorithms intended for broad use in contrast-enhanced CTs.

## Introduction

Cardiovascular disease remains the leading cause of death in the United States [1]. Approximately 50% of these deaths have no prior clinical symptoms or diagnoses [1, 2]. Abdominal aortic calcification shows strong promise as a clinical correlate and an effective predictive tool of future cardiovascular events [3–7]. Early risk identification is important to discern the type and intensity of preventative interventions; 50–90% of cardiovascular events are estimated to be preventable [8, 9]. Coronary artery calcium (CAC) scores paired with cardiovascular risk scores have traditionally provided the strongest estimates for cardiovascular event risk, however, performing coronary calcium scoring in asymptomatic individuals is not encouraged [2, 10]. Ongoing model development efforts attempt to leverage the large number of existing abdominal computed tomography (CT) scans to opportunistically identify and quantify abdominal aortic calcification to better estimate population-level risk, particularly among those with asymptomatic or sub-clinical disease who do not meet criteria for CAC scoring [5, 6, 11]. A substantial obstacle facing the development of robust abdominal aortic calcification models from CT is the confounding created by the the use of intravenous contrast, specifically the pixel attenuation heterogeneity between contrast-enhanced and non-contrast scans [12–14].

Intravenous contrast media is used to increase the radiodensity of vascularized anatomy to better visualize and differentiate that anatomy from adjacent tissues having otherwise similar radiodensities [15]. Its precise effect on CT pixel attenuation in the aorta is complex, determined by multiple scan protocol factors—bolus volume, concentration, injection rate, IV location and bore, and timing to and duration of image acquisition—which are then modulated by the individual physiology of the patient as the bolus mixes in the bloodstream [16]. The use of contrast media is tempered by toxicity concerns due to its high iodine concentration, and is often avoided entirely in patients with reduced kidney function [15, 16]. Furthermore, excess contrast media enhancement in localized regions of the body may cause beam-hardening and scatter artifacts which can obscure anatomy of interest. Finally, most CT scanning protocols are tuned to balance competing goals including minimizing radiation exposure, while maximizing visibility of particular details and spatial, temporal, and contrast resolution. Understandably, the goal of inter-scan homogeneity for the purpose of downstream image processing algorithms often falls secondary to these goals.

Calcified plaque has a higher radiodensity than blood or vessel walls, making it readily segmented on non-contrast scans using traditional thresholding techniques at established threshold levels of 130 HU. However, on contrast-enhanced studies this task becomes increasingly difficult when the adjacent iodine density reaches the same Hounsfield Unit (HU) range as the calcium being segmented [17].

Recent studies have raised the threshold level to 250 HU when detecting calcification on delayed-phase contrast scans and then related (via linear regression) the resulting Agatston scores to scores derived from non-contrast scans [17]. However, we were unable to find research exploring either (1) appropriate calcification thresholds to use in more highly enhanced (venous or arterial-phase) CT scans, or (2) volumetric loss in detectable calcification when using different thresholds.

In this study we target those knowledge gaps by reporting aorta attenuation data from different patient populations under a variety of contrast-enhanced CT acquisitions. To aid future efforts in automatic calcification scoring, we also examine the practical effects of identifying calcified plaque regions via increasingly higher HU thresholds as the baseline radiodensity of the aorta rises due to the presence of contrast medium in the vessel.

## Materials and methods

The CT scans used in this study were obtained from Michigan Medicine, investigation and analysis has Institutional Review Board approval (HUM-00041441). All data was anonymized prior to review and informed consent was waived due to the retrospective nature of the data. All image processing and data analysis was performed in MATLAB [18].

### Patient populations

A total of 6319 patients from three already existing institutional patient cohorts were included in this study. For each patient cohort one or more CT acquisitions were performed using a variety of contrast enhancement protocols described below. No participants were excluded from the study. Patient counts, demographic breakdowns, and CT acquisition settings are summarized in Table 1. Fig 1 visualizes the sample sizes, and scan acquisition availability for cohort participation.

One cohort of patients are kidney donors who underwent assessment CT scanning prior to donation, and have been previously described [19]. These patients were scanned using a non-contrast acquisition (KD-NC) and/or an arterial-phase contrast-enhanced acquisition (KD-Art). The second patient cohort underwent diagnostic scanning for pancreatic cancer under one or more of the following acquisitions: no contrast (PC-NC), arterial-phase (PC-Art), venous-phase (PC-Ven), or delayed-phase (PC-Del). The third patient cohort

**Table 1. Patient demographics, scan acquisitions, and vertebral levels within scan limits.**

| Key | Kidney Donors (n = 3078) | | Pancreatic Cancer (n = 188) | | | | Trauma (n = 3065) | |
|---|---|---|---|---|---|---|---|---|
| | KD-NC | KD-Art | PC-NC | PC-Art | PC-Ven | PC-Del | TR-1 | TR-2 |
| Contrast phase | None | Arterial | None | Arterial | Venous | Delayed | Mixed | Mixed |
| Number (n) | 3042 | 2743 | 179 | 78 | 179 | 28 | 2454 | 611 |
| Age μ±SD (years) | 41.4±12.0 | 41.5±12.1 | 62.8±10.9 | 61.8±11.6 | 63.4±11.0 | 62.6±11.3 | 42.0±20.7 | 44.2±18.8 |
| Age range (years) | 18–76 | 18–76 | 34–86 | 34–83 | 34–86 | 42–82 | 16–90 | 17–94 |
| Men (n (%)) | 1166 (38%) | 1036 (38%) | 97 (54%) | 45 (58%) | 97 (54%) | 14 (50%) | 1530 (62%) | 449 (73%) |
| Women (n (%)) | 1876 (62%) | 1707 (62%) | 82 (46%) | 33 (42%) | 82 (46%) | 14 (50%) | 924 (38%) | 162 (27%) |
| Acquisitions (n) * | 3062 | 2803 | 179 | 78 | 179 | 28 | 3384 | 612 |
| Vertebral obs. (n) | 17221 | 16857 | 1264 | 595 | 1520 | 206 | 21816 | 5829 |
| Slice Thickness (mm) | 5 | 0.625–2.5 | 5 | 2.5–5 | 5 | 2.5–5 | 0.625–5 | 5 |

* TR-1 trauma scans often obtained chest/abdomen regions as separate CT acquisitions. A small number of subjects from other cohorts underwent repeat scanning.

All acquisitions were obtained at 120 kVp with a reconstruction kernel optimized for viewing soft tissue.

Patient weight information was only available in KD and TR.

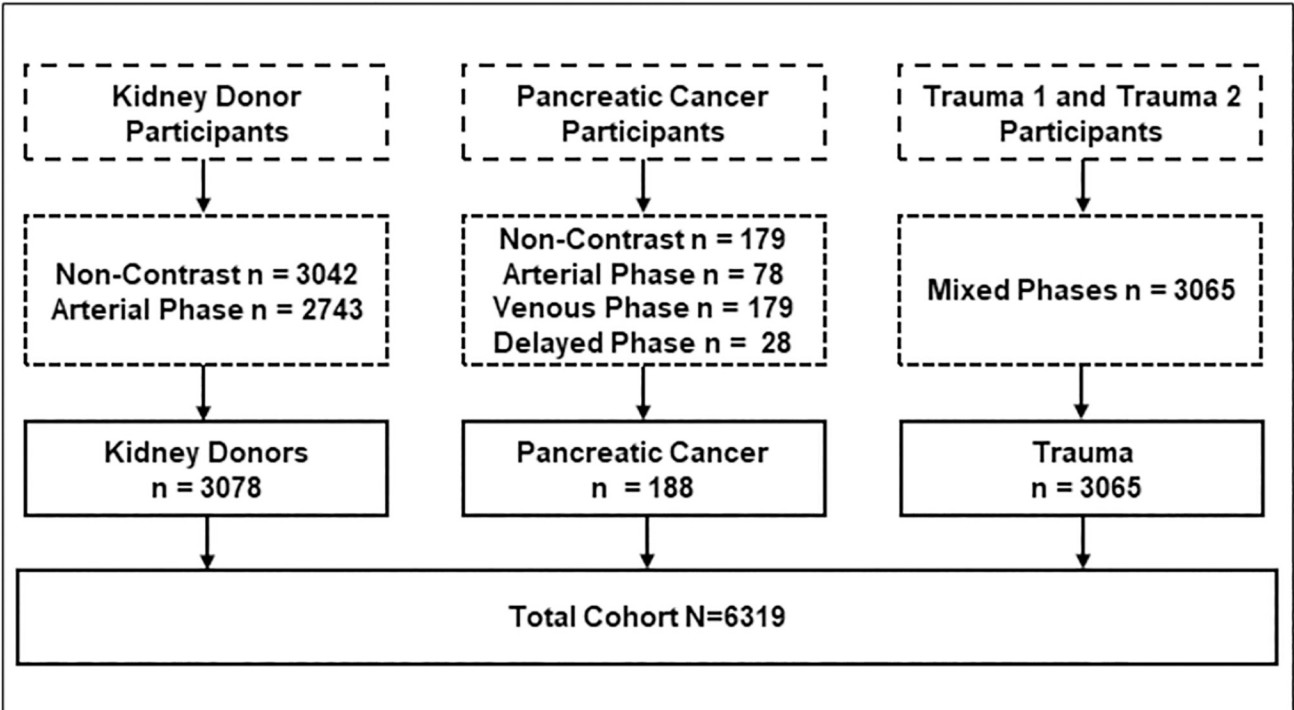

**Fig 1. Flow chart describing pooled cohorts and scan cohort scan phase heterogeneity.**

received diagnostic CT scans as part of trauma care and are described elsewhere [20, 21]. These trauma cohorts (TR-1 and TR-2) each underwent CT scanning with mixed protocols in terms of their contrast enhancement. Due to the high variability of a patient's positioning, hemodynamic status and IV location in a trauma setting the bolus of intravenous contrast media administered is non-uniform across patients leading to a wide range of aortic enhancement for any given CT protocol. In a small fraction of patients (particularly those with reduced kidney function), contrast injection may be foregone entirely.

Details of the contrast bolus itself were either simplified (e.g., simply 'Contrast applied' or 'IV'), or missing entirely from the DICOM headers of the majority of scans and were not considered for this study.

## CT image processing

A combination of convolutional neural networks (CNNs) and post-processing were used to automatically segment aorta geometry on all scans. First, an ensemble of axial, sagittal, and coronal semantic segmentation CNNs were run on full-slice images to produce a probability volume identifying the descending aorta between the arch and femoral bifurcation. A post-processing step then fitted a smoothed spline along the regional maxima in this volume to represent the centerline geometry of the aorta. A follow-up semantic segmentation CNN was then run on small re-sliced patches that were taken along a series of planes perpendicular to, and centered on, that aorta centerline geometry. This second phase, focused only on the aorta, allowed for a more detailed 3D aorta segmentation that was robust to variation in anatomy seen throughout the rest of the scan. Finally, a body region regression CNN was run that associated each slice with a vertebral level along the spine (L4, L3, etc.) by proximity of the slice

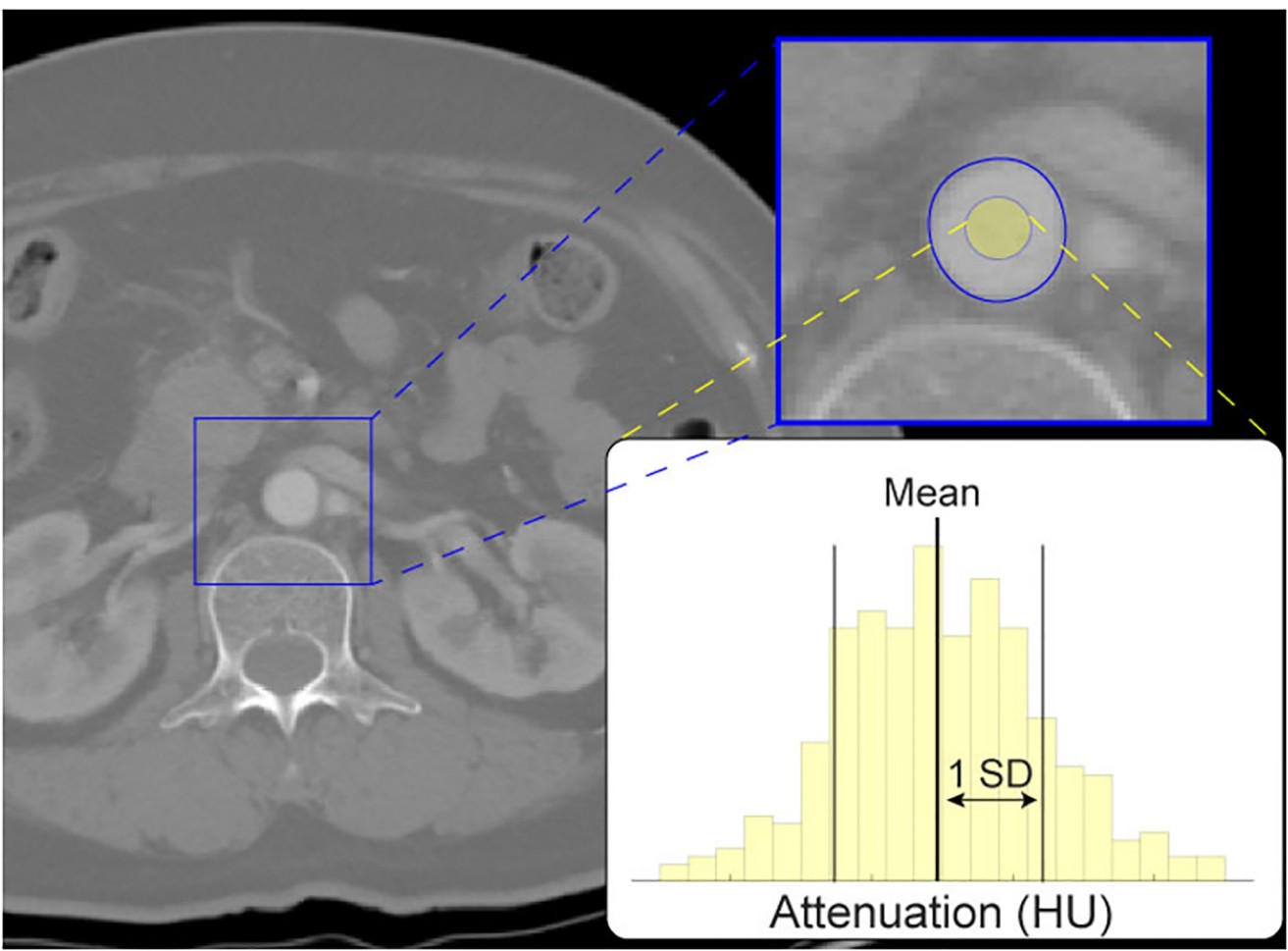

**Fig 2. Aorta attenuation (AoHU) is characterized by the mean and standard deviation of pixels inside a central aorta region.**

location to midpoints of each vertebral body. Slices were grouped by vertebral level, and both partial vertebrae (i.e., those not having both end plates fully within the scan extent) and vertebrae outside the range L4 to T8 were excluded.

A central region within the aorta was identified on every image slice by eroding the segmented aorta to 50% of its original radius (Fig 2). The mean and standard deviations (SD) of all pixel values both within this central zone and belonging to each vertebral level were calculated for all scans.

Summary images from each vertebral observation were reviewed for adequate segmentation of a central zone that excluded regions of calcification. 26 scans from the TR-1 trauma cohort were excluded due to severe disruption of aorta anatomy, the presence of aortic stents or other medical devices, or automatic algorithm failure.

## Statistical analysis

**Associations with AoHU.** Difference in mean aorta attenuation (AoHU) at L1 between males and females across patient sub-populations were assessed using two-tailed t-tests assuming unequal variance.

Within each sub-population the univariate association between demographic and scan predictors (age, weight, slice thickness) and AoHU were assessed via simple linear regression. AoHU associations to vertebral level were assessed per sub-population via simple linear regression using observations from all vertebrae and treating levels (i.e., L4 = 1, L3 = 2, etc.) as continuous.

**Calcification threshold analysis.** The appropriateness of different HU thresholds for detecting plaque pixels was assessed by estimating population-wide averages in two types of plaque detection error (false-positive and false-negative pixel errors, see Fig 3 for illustration) as a function of changing threshold level.

False-negative errors were estimated based on a distribution of plaque pixel intensity values obtained from plaques identified within a representative sample of 50 randomly selected non-contrast scans (25 each from males and females in the KD-NC cohort). Plaque regions comprising this distribution were segmented at a uniform threshold of 130 HU and reviewed visually prior to inclusion. Taking the assumption that contrast agent does not alter the attenuation of calcified plaque pixels themselves (explored further in the discussion), the proportion of this distribution falling below a given HU value indicates the expected false-negative rate incurred from all scans when thresholding for calcification at that level.

False-positive errors were estimated via the assumption that pixel attenuation of non-plaque pixels within aorta regions is normally distributed, such that any observed mean/SD AoHU values also indicate the proportion of that vessel's area or volume exceeding a given HU threshold.

Average rates of false-positive and false-negative pixel classification errors were calculated under a uniform threshold scheme (i.e, where errors were assessed with all scans using the same single threshold) and a dynamic threshold scheme where scans were assessed using individually-calculated threshold values of $N$ standard deviations above the mean. Results were

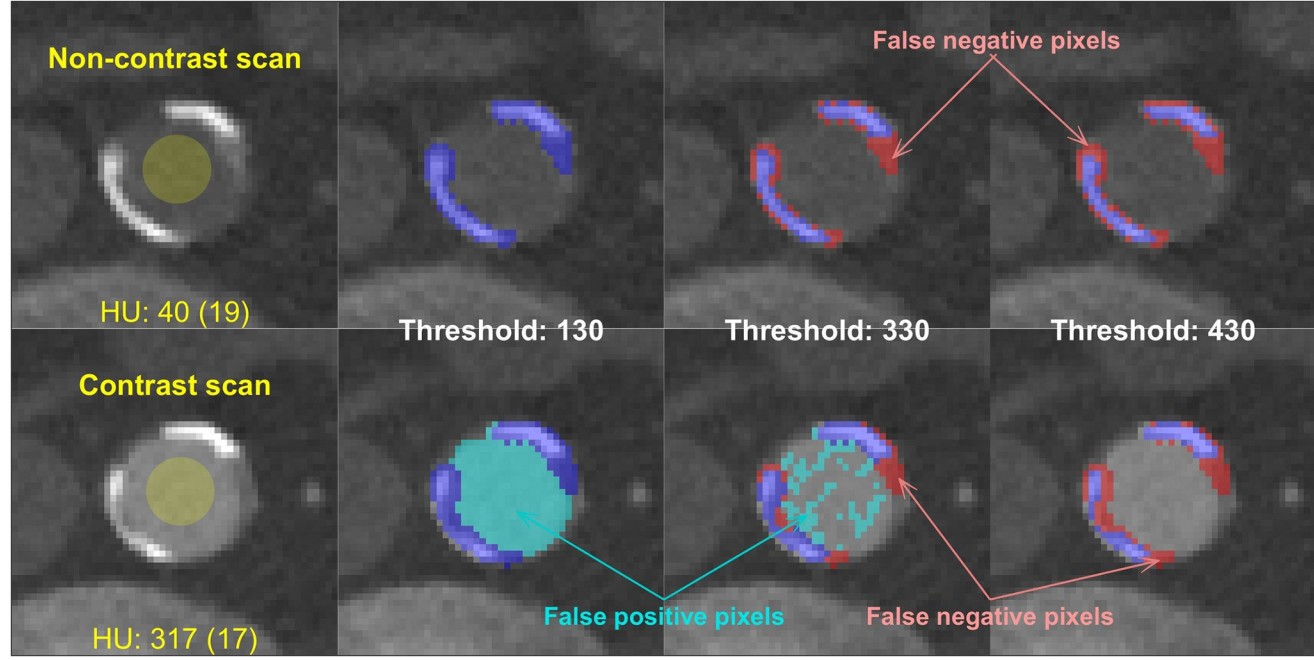

**Fig 3. Example pixel misclassification types when detecting calcified plaque via HU thresholds.** Contrast-enhanced scans have higher baseline HU within the aorta resulting in false positive errors at low thresholds. Higher HU thresholds reduce false positive errors but can produce false negative errors for plaque pixels of moderate density.

aggregated separately by sex and cohort (arterial, venous, and delayed-phase contrast enhancement groups).

## Results

### Aorta CT attenuation

Distributions in aorta attenuation (mean AoHU from the L1 vertebral level) separated by sex and cohort are summarized in Fig 4. Other vertebral levels L4 through T9 showed similar distributions. Mean AoHU for females was significantly higher than for males in all contrast-enhanced cohorts ($p < 0.001$ in all comparisons). On average, female mean AoHU was higher by 20 HU in delayed-phase scans, 22 HU in venous scans, 48 HU to 65 HU in arterial scans, and 27 HU to 28 HU in the mixed-enhancement trauma scans. This effect was not present in non-contrast scan cohorts (average difference less than 0.4 HU, $p > 0.05$). After adjusting for subject weight, AoHU in arterial-phase scans remained higher in women than in men by a difference of 17 HU ($p < 0.001$).

Regression coefficients of mean AoHU versus subject age, weight, and vertebral level are summarized in Table 2. Age was associated with AoHU within some individual cohorts, but the size and directions of association were mixed across groups and the explanatory power

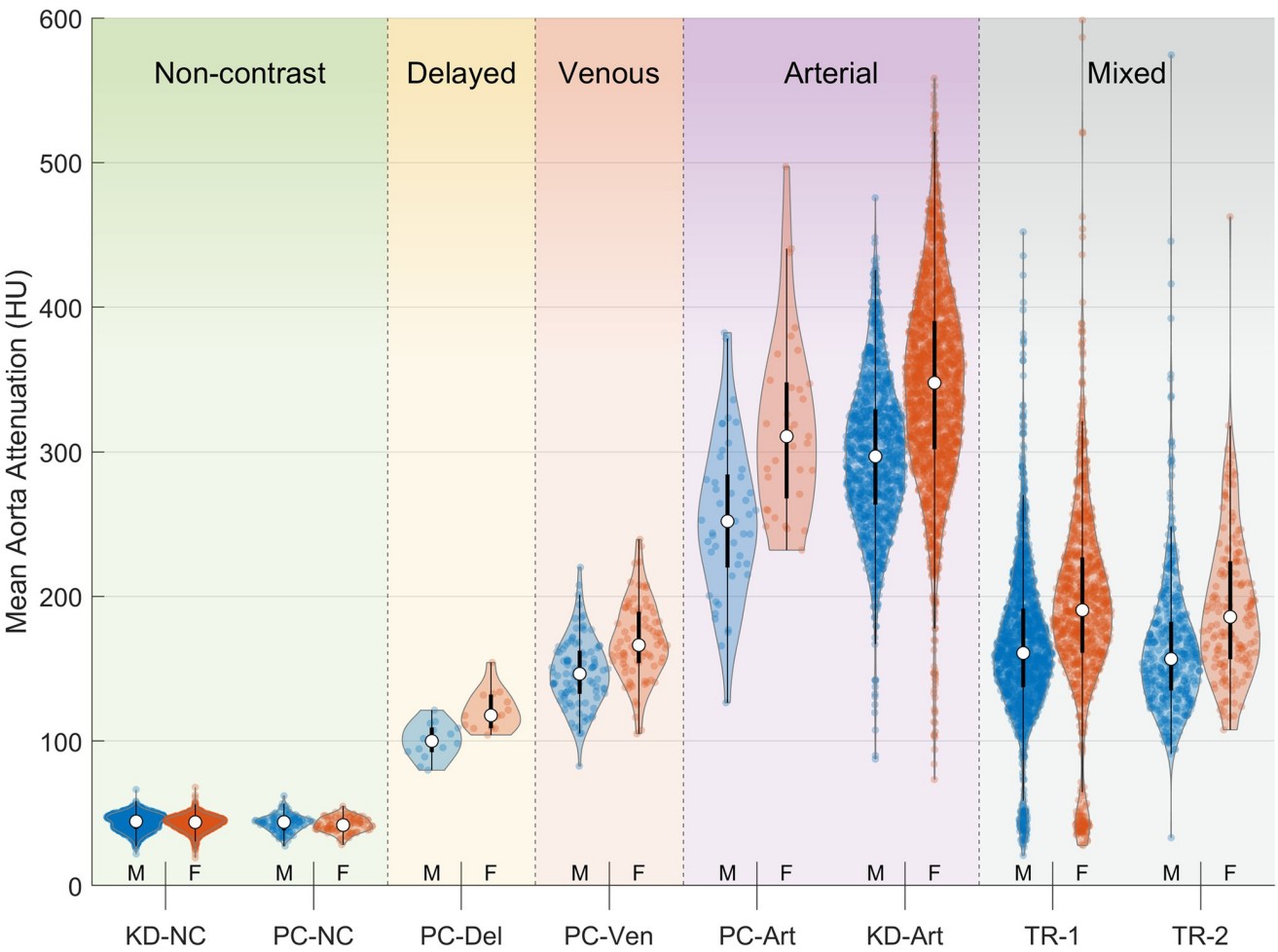

**Fig 4. Distribution of mean AoHU at L1 for males and females within each contrast enhancement group.**

**Table 2. Simple linear regression coefficient (Est.) and fit statistic ($R^2$) for mean AoHU versus demographic and scan factors.**

| | Age (HU/decade) | | | | Weight (HU/10kg) | | | | Vertebral level (HU/vertebra) | | | | Slice Thickness (HU/mm) | | | |
| | M | | F | | M | | F | | M | | F | | M | | F | |
| Cohort | Est. | $R^2$ | Est. | $R^2$ | Est. | $R^2$ | Est. | $R^2$ | Est. | $R^2$ | Est. | $R^2$ | Est. | $R^2$ | Est. | $R^2$ |
|---|---|---|---|---|---|---|---|---|---|---|---|---|---|---|---|---|
| KD-NC | -0.1 ** | 0.09 | -0.1 ** | 0.07 | -0.1 ** | 0.11 | -0.1 ** | 0.16 | -1.7 ** | 0.15 | -1.6 ** | 0.14 | | | | |
| PC-NC | -0.0 | -0.01 | -0.0 | -0.01 | | | | | -0.6 | 0.00 | -0.9 ** | 0.09 | | | | |
| PC-Del | 0.1 | -0.08 | -0.3 | -0.01 | | | | | -1.9 ** | 0.13 | -1.0 | 0.02 | | | | |
| PC-Ven | 0.6 * | 0.07 | 0.1 | -0.01 | | | | | -1.0 * | 0.01 | -1.0 | 0.01 | | | | |
| PC-Art | 1.3 | 0.06 | -1.8 | 0.07 | | | | | -2.5 | 0.01 | -1.8 | 0.00 | -8.7 | 0.02 | -16.4 | 0.05 |
| KD-Art | 0.5 ** | 0.01 | 1.1 ** | 0.03 | -1.6 ** | 0.23 | -2.3 ** | 0.24 | -6.0 ** | 0.04 | -7.1 ** | 0.03 | -5.9 * | 0.01 | -13.0 ** | 0.03 |
| TR-1 | -0.0 | -0.00 | 0.3 * | 0.01 | -0.9 ** | 0.11 | -1.0 ** | 0.06 | 5.7 ** | 0.06 | 6.6 ** | 0.05 | -5.2 ** | 0.01 | -6.5 * | 0.01 |
| TR-2 | 0.6 ** | 0.05 | 0.6 * | 0.04 | | | | | -0.9 ** | 0.00 | -0.8 | 0.00 | | | | |

\* $p < 0.01$

\*\* $p < 0.001$

(adjusted $R^2$) of the age coefficient was less than 9% in all cohorts. Increased subject weight was significantly associated with a reduction in AoHU in all cohorts with available weight data (kidney donors and TR-1), and explained between 6% to 24% of AoHU variation within cohorts at a rate of up to -2.3 HU per 10 kilogram in arterial-phase KD-Art CTs. Vertebral level was also associated with AoHU in some cohorts, with strongest effects being an increase in attenuation of between 1 HU to 2 HU per vertebral level moving inferiorly down the spine for non-contrast scans and 2 HU to 7 HU for arterial-phase scans. This effect had most explanatory power in non-contrast groups ($R^2$ up to 15%) compared to contrast-enhanced cohorts ($R^2$ between 1% to 12%). Higher CT slice thickness generally saw minor reductions in mean AoHU, but these changes were not significant in most cohorts or explained little cohort variance ($R^2 < 3\%$).

Standard deviation (SD) within pixels sampled from the central region of the aorta are summarized by sex and cohort in Fig 5. Sex-based differences in AoHU SD were small, with only KD-NC (males 1.9 HU higher) and KD-Art (males 1.4 SD higher) cohorts showing mean differences of more than 1.0 HU. Associations between demographic variables and standard deviation AoHU (Table 3) were weak in most cohorts ($R^2$ less than 3% in almost all cases), with only weight of KD-NC subjects showing substantial effect ($R^2$ up to 40%) at an increase in standard deviation of 1.2 HU per 100 kg increase in weight. Slice thickness in the KD-Art and TR-1 cohorts was associated with reduced pixel variance in the aorta, lowering SD by up to 5.7 HU per 1 mm increase in slice thickness.

## Expected error rates in plaque detection

Pixel attenuation of calcified plaque content detected in the 50 randomly selected KD-NC scans ranged from 130 HU (the applied threshold) to 1400 HU with a median value of 256 HU. Fig 6 shows its overall distribution and cumulative density function, which reflects the proportion of plaque pixels that would no longer be identified as plaque (i.e., false-negative detection errors) if the CT threshold for calcification were set above 130 HU.

Fig 7 shows that same false-negative rate (top left) compared to the (top right) false-negative error rate that would occur if, instead of using a uniform threshold shared across all scans, different thresholds were dynamically and independently applied to each CT image based on the pixels observed within the central region of that aorta. Dynamic threshold error rates are plotted by the number (N) of SDs above the mean density found within the central aorta.

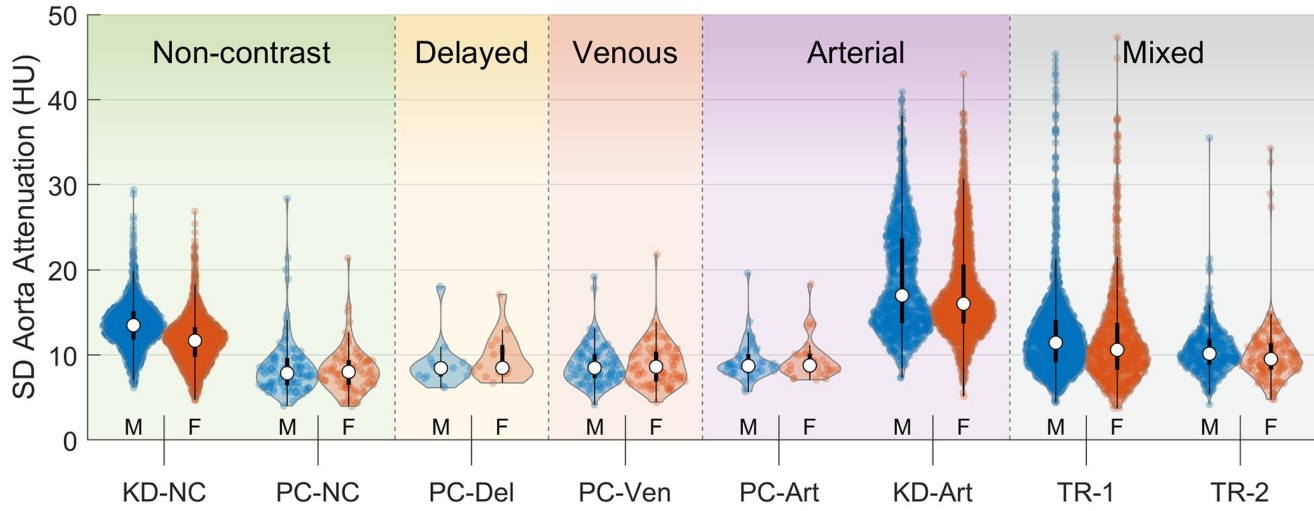

**Fig 5. Distribution of standard deviation in AoHU within the center of the aorta at L1 for males and females within each contrast enhancement group.**

Fig 7 also shows the false-positive detection rates (i.e., the proportion of baseline aorta contents that get falsely identified as calcified plaques) when single plaque threshold values are shared across all scans (bottom left) or when dynamically calculated based on aorta contents (bottom right). For this dynamic threshold paradigm, the proportion of falsely identified plaque pixels within the aorta is a simple property of normal distributions. A dynamic threshold at 3 SDs above the mean AoHU produces just 0.13% false positive pixels within aortas from scans of any contrast level. Due to the high baseline AoHU of arterial scans, that same low false-positive rate is only achieved in the uniform threshold paradigm at levels of 550 HU or above, which in turn correspond to false-negative rates (i.e., missed calcification due to this high threshold) of 85% in all scans. Comparatively, the false-negative rates at a dynamic threshold of 3 SDs are approximately 5%, 25%, and 75% for delayed, venous, and arterial scans, respectively.

**Table 3. Changes in standard deviation AoHU by demographic factors.**

| Cohort | Age (HU/decade) M Est. | $R^2$ | Age F Est. | $R^2$ | Weight (HU/10kg) M Est. | $R^2$ | Weight F Est. | $R^2$ | Vertebral level (HU/vertebra) M Est. | $R^2$ | Vertebral F Est. | $R^2$ | Slice Thickness (HU/mm) M Est. | $R^2$ | Slice F Est. | $R^2$ |
|---|---|---|---|---|---|---|---|---|---|---|---|---|---|---|---|---|
| KD-NC | 0.03 ** | 0.01 | 0.02 ** | 0.01 | 0.11 ** | 0.35 | 0.12 ** | 0.40 | 0.20 ** | 0.01 | 0.11 ** | 0.00 | | | | |
| PC-NC | 0.07 | 0.04 | 0.03 | -0.00 | | | | | 0.59 | 0.00 | -0.14 | 0.01 | | | | |
| PC-Del | -0.08 | 0.03 | 0.09 | 0.07 | | | | | -0.13 | 0.01 | 0.04 | -0.01 | | | | |
| PC-Ven | -0.00 | -0.01 | 0.05 | 0.03 | | | | | -0.14 ** | 0.02 | -0.20 ** | 0.02 | | | | |
| PC-Art | -0.01 | -0.02 | 0.02 | -0.03 | | | | | -0.18 | 0.00 | 0.07 | -0.00 | -0.15 | -0.02 | -0.29 | -0.01 |
| KD-Art | -0.00 | -0.00 | 0.00 | -0.00 | 0.03 * | 0.01 | 0.07 ** | 0.03 | 0.34 ** | 0.01 | 0.21 ** | 0.00 | -5.63 ** | 0.56 | -4.18 ** | 0.45 |
| TR-1 | 0.02 | 0.00 | 0.01 | 0.00 | 0.04 ** | 0.02 | 0.05 ** | 0.03 | 0.55 ** | 0.05 | 0.35 ** | 0.01 | -3.44 ** | 0.37 | -3.16 ** | 0.34 |
| TR-2 | 0.03 ** | 0.03 | 0.03 | 0.01 | | | | | 0.01 | -0.00 | -0.18 ** | 0.01 | | | | |

* $p < 0.01$

** $p < 0.001$

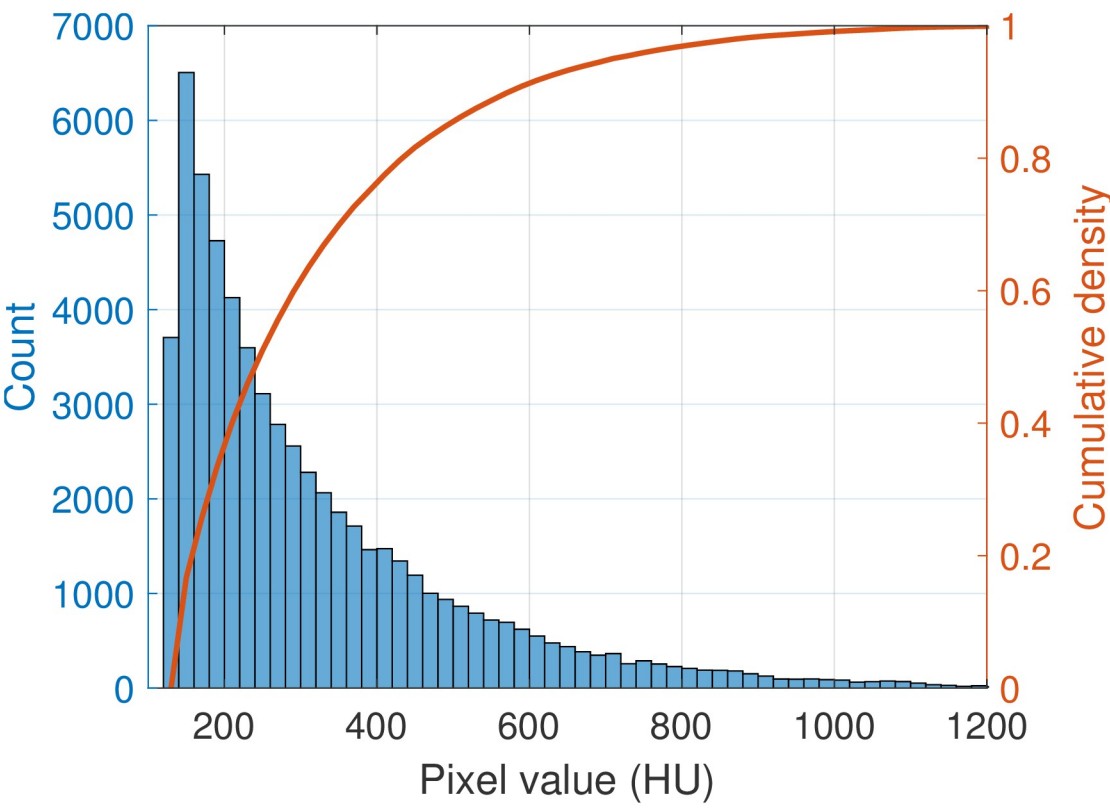

**Fig 6. Distribution (bars) and cumulative density function (line) of calcified plaque pixel attenuation in 50 non-contrast CT scans.** The CDF line reflects the false-negative detection rate as true plaque pixels are excluded at increased CT thresholds.

## Discussion

This analysis evaluated the central vessel HU in multiple large cohorts of aortas scanned with varying degrees of contrast-enhancement. The first of two main findings is that female aortas were consistently enhanced to higher radiodensities than their male counterparts, independent from subject body size. The second finding is that the variation in density introduced from contrast enhancement makes detection of plaque content via uniform HU thresholds an untenable general technique; dynamic thresholds calculated from local aorta content are more robust to scan variation.

Aorta attenuation due to the presence of contrast media can be variable due to confounding from multiple factors. Our cohorts' mean AoHU ranged from 110 ± 16 HU in delayed-phase scans to 326 ± 68 HU in arterial-phase scans. [14] examined the arterial phase acquisition of four-phase liver scans, and histogram figures indicated an average vessel density of approximately 120 HU while a level of 300 HU lay sufficiently far above baseline attenuation of all aortas within their arterial-phase scan cohort to be used as a uniform threshold for plaque content. When comparing options for variations in post-bolus acquisition delays in arterial-phase scans [22], found mean/SD aoHU to be approximately 450 ± 90 HU. Given such disparate literature values, the mean AoHU ranges for our cohorts are certainly compatible with past results. More importantly, the wide-ranging aorta enhancement values come from scans identified as arterial-phase, illustrating that such categories should not be expected to provide homogeneous enhancement even within a single cohort, let alone across institutions or scanning protocols. This reinforces the notion that the most reliable indicator of aorta attenuation

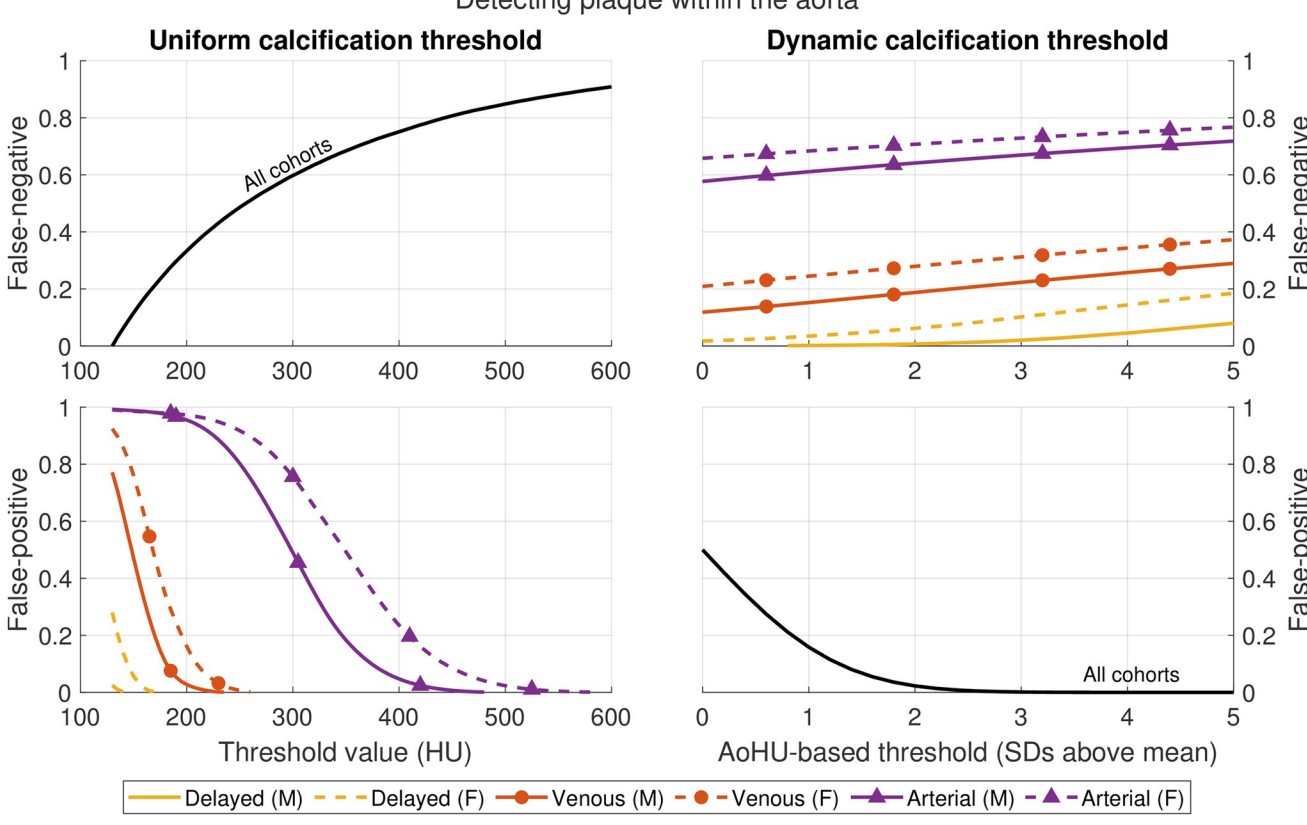

**Fig 7. Estimated rates of false-positive (aorta contents incorrectly identified as plaque) and false-negative (plaque content incorrectly identified as aorta) errors when thresholding pixels within the aorta to detect calcified plaque regions.** A uniform threshold scheme shares the same single threshold across all scans and slices. A dynamic scheme calculates separate thresholds based on pixels observed within the central aorta.

will come from pixels within the aorta itself, rather than any external classification of scan and contrast media bolus parameters.

A primary determinant of vascular enhancement is the bolus volume, concentration, and rate of IV contrast media supplied prior to or during the CT acquisition. Many institutions use a fixed volume of contrast media for abdominal scans, but a growing number of studies suggest weight-based protocols offer more consistent image quality across a range of body weights, and at lower total contrast usage and cost [23–26]. Nevertheless, average blood volumes in women are lower than those in men for a given body size, and results presented here are consistent with past studies finding higher female aorta enhancement (291 ± 49 HU vs. 260 ± 55 HU) that was not accounted for by weight [27–29]. Such findings make lean body mass formulations which include sex-specific terms for contrast media bolus volume a promising alternative [30]. Solutions such as correction universal correction factors for contrast dosing should be considered by experts with multi- and interdisciplinary content knowledge, as well as based in strong and repeatable evidence.

We examined this sex-based bias in aorta attenuation, as well as the expected variation due to different contrast phase imaging, from the perspective of downstream image algorithms employed to evaluate abdominal aortic plaque burden across a wide range of scans. Previously, CNNs have been used to extract aortic plaque regions from contrast-enhanced abdominal CTs, but only for delayed-phase acquisitions in which a threshold of 250 HU was deemed

appropriate to discriminate plaque from aorta contents during training [17]. In that study the target validation metric was an Agatston calcification score prediction obtained from non-contrast scans (rather than, for example, total plaque volume), which was then adjusted by a linear coefficient to counteract a score underestimation which has also been reported in contrast-enhanced coronary artery scans [31]. The reason for this underestimation is that Agatston scoring assigns pixel scores in one of four HU intensity bins (1 for 130 HU to 199 HU, 2 for 200 HU to 299 HU, 3 for 300 HU to 399 HU, and 4 for 400 HU or greater) which are then multiplied by plaque area on a given image. Reductions both in total plaque area and in accumulated bin scores occurs as baseline aorta attenuation exceeds these bin limits, and this relationship may not be linear across the full range of enhancement protocols. Results here (Fig 7) suggest that false-negative loss is indeed non-linear, and a uniform threshold of 250 HU would underestimate plaque pixel volume by approximately 50% compared to the 130 HU used for non-contrast scans. This could be reduced to 10% and 3% for females and males in delayed-phase scans by using a 3 SD dynamic threshold, but false-negative losses in venous or arterial-phase scans would be higher.

When interpreting results from this study, a number of limitations should be considered. Firstly, while the presented data span different cohorts and scan purposes, they do not represent the full breadth of CT scan protocols from which abdominal aortic calcification scoring might ultimately be obtained.

It should also be noted that false-negative and false-positive losses are calculated based on assumptions about pixel attenuation distributions within the aorta and within regions of calcified plaque. Preliminary inspection via linear regression of paired plaque pixel intensities shown in Fig 3 shows strong agreement with contrast-enhanced HU values ($y$) related to non-contrast values ($x$) via $y = 1.1x + 2.7$ (adjusted $R^2 = 0.97$), but further work is required to define this relationship across a wider variety of scans. Similarly, when taken at face value the results in Fig 7 indicate that aortas in arterial-phase scans have such high baseline attenuation that even dynamic thresholds incur a false-negative loss of around 75% of true calcified content. This result, however, reflects a worst-case scenario in which pixel intensity alone is available, and morphology and spatial information is entirely discarded. In reality, most of those lower-intensity plaque pixels come from regions immediately surrounding peaks of higher-intensity plaque pixels that were not part of that false-negative loss (see Fig 3). CNNs and other modern image processing algorithms are well suited to leveraging such spatial information so there is an expectation that large portions of these false-negative losses may be recouped.

Finally, other studies have warned about potential for errors when estimating abdominal aortic calcification scoring using algorithms only validated for coronary artery calcification scoring and with specialized scanning protocols [14, 32]. This current study aims to quantify expected rates for some of these errors, especially as enhancement of the aorta changes due to contrast media. In particular, we wish to promote awareness that without special care, the current sex-based biases in differential vessel attenuation from contrast material can potentially propagate to further unintentional bias in downstream calcification scoring calculations, particularly when uniform HU thresholds are employed. Even in the likely case that future deep learning CNN models replace direct HU thresholding steps during plaque detection, care must still be taken in cases where the models themselves are trained or validated with ground truth data generated from pixel-based thresholding.

## Conclusion

In summary, we have reported distributions of aorta attenuation in different clinical CT scans exhibiting a wide range of contrast-material enhancement, and highlighted a sex-based

difference which may bias downstream calcification scoring. To inform future efforts in automatic calcification scoring algorithms, we also report the plaque detection error rates across the range of contrast-enhanced scan protocols that might be expected under uniform and dynamic HU thresholding schemes.

## Supporting information

**S1 Table. Participant aortic attenuation data.** Includes MAG scan identification number, vertebra index identifier, patient identification number, dataset tag, participant age, participant sex, participant weight (in kg), aortic attenuation mean, aortic attenuation standard deviation.
(CSV)

## Author Contributions

**Conceptualization:** Sven A. Holcombe, Steven R. Horbal, Stewart C. Wang.

**Data curation:** Sven A. Holcombe.

**Formal analysis:** Sven A. Holcombe.

**Investigation:** Sven A. Holcombe, Steven R. Horbal, Brian E. Ross, Edward Brown, Brian A. Derstine, Stewart C. Wang.

**Methodology:** Sven A. Holcombe, Brian A. Derstine.

**Resources:** Brian E. Ross, Stewart C. Wang.

**Software:** Sven A. Holcombe.

**Visualization:** Sven A. Holcombe.

**Writing – original draft:** Sven A. Holcombe, Steven R. Horbal, Brian A. Derstine.

**Writing – review & editing:** Sven A. Holcombe, Steven R. Horbal.

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
