## [Decision Letter · Decision Letter 0]

9 Sep 2022

PONE-D-22-18603Variation in baseline aorta attenuation in contrast-enhanced CT and its implications for calcification thresholdsPLOS ONE

Dear Dr. Horbal,

Thank you for submitting your manuscript to PLOS ONE. After careful consideration, we feel that it has merit but does not fully meet PLOS ONE’s publication criteria as it currently stands. Therefore, we invite you to submit a revised version of the manuscript that addresses the points raised during the review process.

We look forward to receiving your revised manuscript.

Kind regards,

Jianhong Zhou

Staff Editor

PLOS ONE

Journal Requirements:

Reviewers' comments:

Reviewer's Responses to Questions

**Comments to the Author**

1. Is the manuscript technically sound, and do the data support the conclusions?

Reviewer #1: Yes

Reviewer #2: Yes

2. Has the statistical analysis been performed appropriately and rigorously? 

Reviewer #1: I Don't Know

Reviewer #2: Yes

3. Have the authors made all data underlying the findings in their manuscript fully available?

Reviewer #1: Yes

Reviewer #2: Yes

4. Is the manuscript presented in an intelligible fashion and written in standard English?

Reviewer #1: Yes

Reviewer #2: Yes

5. Review Comments to the Author

Reviewer #1: General Comments

The authors assessed variation in baseline and contrast-enhanced aorta attenuation in CT and its implications for calcification thresholds. They found that uniform CT thresholds for calcified plaques incur high rates of pixel classification errors in contrast-enhanced scans, which can be minimized using dynamic thresholds based on local aorta attenuation. In addition, they found that females had higher aorta attenuation than males in contrast-enhanced scans but not in non-contrast scans and that weight was negatively correlated with aorta attenuation. The main strength of this study is that imaging tools, which allow risk assessment for cardiovascular diseases, could reduce cardiovascular events due to timely reduction of risk factors and timely onset of therapy. The main weakness is that some findings of this study are already well known and that there is no information on how the model introduced by the authors (dynamic thresholds based on local aorta attenuation) influenced cardiovascular risk prediction in their patient population.

Specific Comments

1. Title. As also variation in attenuation in contrast-enhanced CT and not only in baseline aorta attenuation was assessed the title should be adapted.

2. Materials and Methods. Patient Populations. Clear inclusion and exclusion criteria should be provided, additionally visualized with a flow chart.

3. Materials and Methods. Calcification Analysis. This section is confusing. It is not clear how false-negatives and false-positives were assessed. Parts of this section should also be included in the Patient population section.

4. Results. Information on how dynamic thresholds based on local aorta attenuation improved cardiovascular risk prediction compared to uniform CT thresholds in the study population would be very interesting.

Reviewer #2: Thank you for this interesting manuscript, which raises optimal contrasting in different contrast phases to an epidemiologic level. Interesting general observations were made, such as that women generally require a smaller amount of contrast than men and that the formulas used today do not ideally reflect the lean body weight that should be used. Here, I would suggest that you propose a factor, based on your data, of what percentage less contrast agent could be used in women compared to men. In addition, the issue of automatic plaque classification is very important because it can be used to derive prognostic values that can be applied to individuals as part of risk stratification and the next step of risk prevention. Certainly, there has been similar work in the past, but this appears to be of high professional quality.

Abstract. OK

Introduction: may be shortened, but ok

Results: ok

Diskussion: ok

Conclusion: justified, ok

6. PLOS authors have the option to publish the peer review history of their article (what does this mean?). If published, this will include your full peer review and any attached files.

Reviewer #1: No

Reviewer #2: **Yes: **PD Dr. Christoph Artzner

---

## [Author Response · Author response to Decision Letter 0]

19 Sep 2022

We have reviewed the journal requirements and believe they are consistent with those outlined by PLOS One. Our file names have been renamed. Thank you.

The IRB approval was included, and we agree was not thorough enough. We have added the additional sentence and reflects the situation. “All data was anonymized prior to review and IRB review included the waiving of informed consent due to the retrospective nature of the data”.

We have included a caption for the supporting information file at the end of the manuscript. 

Reviewer #1: General Comments

The authors assessed variation in baseline and contrast-enhanced aorta attenuation in CT and its implications for calcification thresholds. They found that uniform CT thresholds for calcified plaques incur high rates of pixel classification errors in contrast-enhanced scans, which can be minimized using dynamic thresholds based on local aorta attenuation. In addition, they found that females had higher aorta attenuation than males in contrast-enhanced scans but not in non-contrast scans and that weight was negatively correlated with aorta attenuation. The main strength of this study is that imaging tools, which allow risk assessment for cardiovascular diseases, could reduce cardiovascular events due to timely reduction of risk factors and timely onset of therapy. The main weakness is that some findings of this study are already well known and that there is no information on how the model introduced by the authors (dynamic thresholds based on local aorta attenuation) influenced cardiovascular risk prediction in their patient population.

Thank you for thorough review of our manuscript. While we agree that some of this information is known, this work provides a mechanism by which the individualized aortic attenuation determines the calcification threshold for more accurate identification of calcification, not necessarily risk prediction in the patient population. The ability to compare heterogenous scans in various contrast phases are important towards harmonizing cardiovascular cohorts, regardless of contrast status, as well as building individualized, unbiased, cardiovascular risk prediction models,

Specific Comments

1. Title. As also variation in attenuation in contrast-enhanced CT and not only in baseline aorta attenuation was assessed the title should be adapted.\\

Please see that we have removed the word ‘baseline’ from the title. 

The "baseline” here is intended to delineate the vessel attenuation within the subluminal space wherein contrast material may directly influence attenuation (as opposed to the lumen or other regions of the vessel wall which may accumulate calcification). We chose this term rather than the less accessible “subluminal attenuation” since our target audience includes algorithm developers as much as clinical professions.

2. Materials and Methods. Patient Populations. Clear inclusion and exclusion criteria should be provided, additionally visualized with a flow chart.

We have added a flow chart to clear up the inclusion criteria—these include 3 already existing cohorts that we have cited more in-depth descriptions of in other published work. We also added “already existing” to the first sentence of the patient population paragraph. No exclusion criteria were utilized. We hope this clears up any potential confusion.

3. Materials and Methods. Calcification Analysis. This section is confusing. It is not clear how false-negatives and false-positives were assessed. Parts of this section should also be included in the Patient population section.

Fair comment. To reduce confusion, we have retitled this section “Calcification threshold analysis”. Please note that our paper does not assess false-negatives and false-positives of, say, vascular disease or other clinical outcome which might be tabulated at the patient level. Instead, this paper analyses false-negative and false-positive pixels that would be identified (or not) as being part of a calcification site on any given scan. Here, the ground truth against which false positive/negatives are assessed is simply the current established standard threshold of 130 HU when applied to a non-contrast scan. If a given subject has an unusually high attenuation within their aorta (typically due to contrast media) then this established threshold is likely to produce false-positive pixels exceeding this threshold that get identified as calcified plaques by a naïve automatic algorithm. We measure the percentage of their vessel which would (erroneously) be labeled as such.

4. Results. Information on how dynamic thresholds based on local aorta attenuation improved cardiovascular risk prediction compared to uniform CT thresholds in the study population would be very interesting.

While we certainly agree this would be ideal, accurate CVD diagnosis for the study population is currently unavailable. We can assure you that there are initiatives to provide greater data interoperability solutions, as well as expert diagnostic mechanisms are being developed by our lab and our working partners for future research capabilities.

Reviewer #2: Thank you for this interesting manuscript, which raises optimal contrasting in different contrast phases to an epidemiologic level. Interesting general observations were made, such as that women generally require a smaller amount of contrast than men and that the formulas used today do not ideally reflect the lean body weight that should be used. Here, I would suggest that you propose a factor, based on your data, of what percentage less contrast agent could be used in women compared to men. In addition, the issue of automatic plaque classification is very important because it can be used to derive prognostic values that can be applied to individuals as part of risk stratification and the next step of risk prevention. Certainly, there has been similar work in the past, but this appears to be of high professional quality.

Abstract. OK

Introduction: may be shortened, but ok

Results: ok

Diskussion: ok

Conclusion: justified, ok

Thank you for the flattering assessment of our work. Given the scope of our expertise, we do not believe that we are in an appropriate position to make recommendations for contrast agent dosing. We were surprised to find this higher aortic enhancement in females relative to males but was only obvious after operationalization of the dynamic threshold concept. Given the biases in our study population, as well as institutional protocols, universal recommendations may be unsuitable. As this work is relatively novel, we hope to inspire other researchers to investigate and reproduce these finding in similar patient-serving environments. These may include inter- and multidisciplinary solutions in general medicine, radiology (and nuclear medicine), computer science, biomedical engineering, and other applicable fields. Hopefully, this work generates interest in highlighting this disparity and influences proper solutions. We have added a few sentences to the discussion to encourage further work.

---

## [Editor Report · Decision Letter 1]

20 Oct 2022

Variation in aorta attenuation in contrast-enhanced CT and its implications for calcification thresholds

PONE-D-22-18603R1

Dear Dr. Horbal,

We’re pleased to inform you that your manuscript has been judged scientifically suitable for publication and will be formally accepted for publication once it meets all outstanding technical requirements.

Kind regards,

Helmut Schoellnast

Guest Editor

PLOS ONE
---

## [Editor Report · Acceptance letter]

2 Nov 2022

PONE-D-22-18603R1 

Variation in aorta attenuation in contrast-enhanced CT and its implications for calcification thresholds 

Dear Dr. Horbal:

I'm pleased to inform you that your manuscript has been deemed suitable for publication in PLOS ONE. Congratulations! Your manuscript is now with our production department. 

Kind regards, 

on behalf of

Dr. Helmut Schoellnast 

Guest Editor

PLOS ONE